# Prevalence and correlates of depressive symptoms among gay, bisexual and other men who have sex with men in the PROUD randomised clinical trial of HIV pre-exposure prophylaxis

Ada Miltz [1], Fiona Lampe,[1] Sheena McCormack,[2] David Dunn,[2] Ellen White,[2] Alison Rodger,[1] Andrew Phillips,[1] Lorraine Sherr,[1] Ann K Sullivan,[3] Iain Reeves,[4] Amanda Clarke,[5] Mitzy Gafos[6]

For numbered affiliations see end of article.

**Correspondence to**
Dr Ada Miltz;
Ada.Miltz.11@ucl.ac.uk

## ABSTRACT

**Objectives** The aim of this analysis is to: (i) assess the prevalence of clinically significant depressive symptoms at baseline and follow-up for participants in the PROUD trial of HIV pre-exposure prophylaxis (PrEP), examining changes in prevalence over time and (ii) investigate the association of socioeconomic and psychosocial factors with depression.

**Methods** PROUD was an open label randomised trial evaluating the benefit of PrEP for 544 HIV-negative gay, bisexual and other men who have sex with men (GBMSM) in England. Enrolment was between 2012 and 2014, with at least 2 years follow-up. Prevalence of depression (score $\geq 10$ on Patient Health Questionnaire-9) was assessed and compared across time-points (using McNemar's $\chi^2$ tests) and between trial arms (using $\chi^2$ tests). Cross-sectional associations with socioeconomic and psychosocial factors were examined using baseline data in modified Poisson regression models and combined 12 and 24 month follow-up data in generalised estimating equations (GEEs). Prevalence ratios (PRs) were presented as unadjusted PR and adjusted PR (aPR) for age, UK birth, sexual identity, university education, London study clinic site and calendar time (and follow-up time-point in GEEs).

**Results** Depression increased significantly from baseline (9.1%; 49/540) to the 12 month (14.4%; 59/410) and 24 month (14.4%; 48/333) follow-ups, possibly explained by underreporting at baseline. The prevalence of depression did not differ by study trial arm, at any time-point. In the baseline analysis, younger age, unemployment and crystal methamphetamine use, was associated with depression. In combined analysis of 12 and 24 month data, measures of intimate partner violence (IPV) (lifetime IPV victimisation aPR 2.57 (95% CI: 1.71 to 3.86)), internalised homophobia (aPR 1.91 (95% CI: 1.29 to 2.83)) and concealment of sexual identity (aPR 1.75 (95% CI: 1.16 to 2.65)), were strongly associated with depression.

**Conclusions** There is a high concomitant burden of psychosocial factors with depression among GBMSM.

**Trial registration number** ISRCTN (ISRCTN94465371) and ClinicalTrials.gov (NCT02065986).

## Strengths and limitations of this study

► This is the first UK study to collect data on depression prevalence over time among gay, bisexual and other men who have sex with men (GBMSM).
► This is the first UK study to investigate associations between a range of psychosocial factors and depression among GBMSM.
► Since the data were analysed cross-sectionally, the direction of associations may be unclear for some factors.
► This study may have lacked power to detect the presence of some associations.
► There is a possibility of unmeasured confounding in this study, as information on childhood sexual abuse, income and levels of social support was not collected.

## INTRODUCTION

There is consistent evidence that depression prevalence is elevated among gay and bisexual men compared with their heterosexual counterparts in high-income countries.[1] Two theoretical models have put forward explanations for the elevated prevalence observed among sexual minorities: minority stress theory and syndemic theory. Minority stress theory describes the psychosocial consequences of being in continual conflict with a discriminatory and heteronormative social environment. Perpetual negative feedback from others is thought to lead to a process of self-stigmatisation termed internalised homophobia, whereby antigay social values/attitudes are directed towards the self. Internalised homophobia often results in deep conflict and poor self-regard with negative consequences such as pervasive expectations of rejection in one's life and

concealment of one's sexual identity. Concealment adds to mental distress by disallowing individuals to affiliate with people of the same sexual identity.[2] Two US studies of gay, bisexual and other men who have sex with men (GBMSM) found strong associations between perceived discrimination on the basis of sexual orientation and depressive symptoms.[3 4] Internalised homophobia may explain this association.

Syndemic theory proposes that concomitant widespread use of recreational drugs and experiences of intimate partner violence (IPV) among sexual minority men may be to blame for elevated levels of depression, as causal, and bidirectional, relationships are postulated between drug use, IPV and depression. The synergistic interaction of these co-occurring factors is thought to result in an exaggerated risk of poor health outcomes.[5] A number of US studies have found strong associations of recreational drug use[3 6 7] and IPV[8] with depressive symptoms. Recently, studies have also found a link between chemsex and depression.[9–11] Chemsex is a cultural phenomenon among a subgroup of gay identified men, which was first described in the UK.[12] It is the intentional use of psychoactive substances (usually one or more of mephedrone, gamma-hydroxybutyrate/gamma-butyrolactone (GHB/GBL) and methamphetamine) to stimulate sexual arousal, facilitate different sexual practices and prolong sexual episodes.

A number of UK studies have examined the prevalence of, and factors associated with, depressive symptoms among GBMSM.[13–18] In several of these studies however, power to assess associations was limited due to small sample size (<200 GBMSM),[13 15 16] and one limitation of the larger studies is that data were not collected on possible psychosocial risk factors (internalised homophobia and IPV).[14 17 18] There is a need to better understand correlates of depression among GBMSM in the UK.

This analysis uses data from the PROUD randomised clinical trial, which evaluated the impact of pre-exposure prophylaxis (PrEP) on HIV incidence, and collected data on a range of psychosocial factors. The aim is to: (i) present the prevalence of depressive symptoms at baseline and follow-up, examining changes in prevalence over time and (ii) drawing on minority stress theory and syndemic theory, investigate the association of demographic, socioeconomic and psychosocial factors with depressive symptoms.

## METHODS
### PROUD trial
The PROUD trial was a multicentre, pragmatic open label randomised trial evaluating the benefit of PrEP as part of a package of HIV risk reduction interventions for HIV-negative GBMSM and trans women.[19] Only three trans women enrolled in PROUD, and therefore data cannot be presented separately for trans women.

Routine attenders at 13 genitourinary medicine (GUM) clinics in England were enrolled between November 2012 and April 2014. Volunteers were eligible if they met the following criteria: male sex at birth, aged 18 years or over, HIV-negative on the day of enrolment or in the past 4 weeks, reported condomless anal sex (CAS) in the past 3 months and expected to have CAS again in the next 3 months.[19] Volunteers were randomised 1:1 to an immediate start of daily oral PrEP or a deferred start after 1 year. However, during follow-up, an unexpectedly high incidence of HIV was observed in the deferral arm (9.0 per 100 person-years, 90% CI: 6.1 to 12.8), which led to the decision by the trial steering committee in October 2014 to offer all participants PrEP. Participants had the opportunity to remain in follow-up for at least 2 years.

### Patient and public involvement
The PROUD pilot study team used a number of mechanisms to promote effective and appropriate participant engagement in the research project (http://www.proud.mrc.ac.uk/about/patient-and-public-involvement-ppi/). A number of meetings or discussion forums were hosted throughout the course of the study, in which participants were actively involved in the research process. Participants were informed of these forums via the study team, advertisements in the clinics or the study website. The types of topics participants were involved in included: (i) development of the trial protocol, including questionnaires, participant information sheets and consent forms, (ii) assessing and improving study procedures, recruitment and retention strategies, (iii) assessing and improving public involvement and communication plans and (iv) development of plans for result dissemination and post-study access to study product. Patient and public involvement activities with a small group of PROUD participants (n=13), in the form of a short qualitative survey, was also undertaken as part of this depression project; responses are considered in the discussion.

### Hypothesised relationships and selection of variables
Causal relationships with depressive symptoms were hypothesised for socioeconomic factors (younger age, lower levels of educational attainment, unemployment and having no ongoing relationship) and psychosocial measures (IPV, early age at anal sex debut (a possible marker of childhood sexual abuse), recreational drug use, alcohol use, internalised homophobia and concealment of sexual identity) collected in PROUD. Hypotheses were based on minority stress theory and syndemic theory as well as observed relationships in previous studies, although such relationships have not been proved to be causal.

### Measures
#### Depressive symptoms
The Patient Health Questionnaire (PHQ-9) was used to measure the prevalence of depressive symptoms at baseline and on an approximately annual basis (month 12 and 24). In the PHQ-9, participants are asked to rate how often over the past 2 weeks they have experienced

nine specific problems (anhedonia, feeling hopeless, trouble sleeping, feeling tired, appetite problems, feeling like a failure, trouble concentrating, being slow/restless and suicidal ideation/self-harm).[20] Response options range from 'not at all' (coded as 0) to 'nearly every day' (coded as 3). In this paper, a missing response was coded as 0. Summing responses from each question generates a depressive symptom score. A cut-off point of 10 or greater for the total score indicates a possible clinically significant depressive condition.[20] In a US validation study of primary care and obstetrics/gynaecology patients, the vast majority of patients (93%) with no depressive disorder diagnosed by a (blinded) mental health professional had a PHQ-9 score less than 10, while 88% with a major depression diagnosis had scores of 10 or greater.[20]

The PROUD study team reported that part way through the trial enrolment process a change in PHQ-9 reporting policy was instated, which continued to be utilised throughout follow-up. From April 2013, participants were informed that after self-completing the questionnaire they would be asked to disclose their response to the suicidal/self-harm ideation question on the PHQ-9 ('Thoughts that you would be better off dead, or of hurting yourself in some way') to trial clinical staff. The Trial Steering Committee recommended this change in reporting policy as a result of one man attempting to take his life during the course of the study.

At the baseline visit men were asked by study staff to note down any medication they were taking. All men who specified that they were taking a medication that has been licensed for treatment of depression were identified.

### Intimate partner violence

At month 12 and 24, participants were asked five questions about IPV victimisation and five questions about IPV perpetration, based on the 'Health and Relationships survey'.[18] Each question had the following four response options: never, more than 1 year ago, within the last year with former partner and within the last year with current partner. The five IPV victimisation questions included the following: 'Have you ever felt frightened of the behaviour of a partner', 'Have you ever needed to ask a partner's permission to work, go shopping, visit relatives or visit friends (ie, beyond the usual of being considerate to and checking with a partner)', 'Have you ever been hit, slapped, kicked or otherwise physically hurt by a partner', 'Have you ever been forced to have sex or made to engage in some sexual activity when you did not want to' and 'Have you ever been forced to have sex without a condom when you did not want to'. The five IPV perpetration questions were identical but asked whether the participant had perpetrated these behaviours. A positive response to any one of the five respective questions was considered to indicate lifetime IPV victimisation or lifetime IPV perpetration. A missing response was included in the 'no IPV' category.

### Age at first anal sex

Data on age at anal sex debut with a male were collected at month 12 and 24. Reporting anal intercourse at: (i) 12 years or younger and (ii) 15 years or younger, was investigated. Measure (i) may include experiences of childhood sexual abuse (CSA), however, since the age of the partner and the nature of the encounter was not known, this measure may not reflect or identify all sexual abuse experienced.

### Recreational drug use

At baseline, participants were asked to report whether they had used recreational drugs in the past 3 months. Chemsex drug use was defined as use of mephedrone, GHB/GBL and/or methamphetamine in the past 3 months. It is of note that men were not asked in PROUD whether these drugs had been taken before or during sex. At the 12 month and 24 month questionnaires, participants were asked whether, in the past 3 months, they had engaged in sex after using recreational drugs, which may also be a proxy measure for chemsex.

### Alcohol use

At baseline, a measure of higher risk alcohol consumption was investigated based on the first two questions of the WHO alcohol use disorders identification test for consumption (AUDIT-C) questionnaire.[21] Participants were asked about the frequency of their alcohol consumption in the past 3 months, with response options: never (coded as 0), monthly or less (coded as 1), two or three times a month (coded as 2), once or twice a week (coded as 3), three or more times a week (coded as 4). Participants were also asked about their usual alcohol intake in units on a typical day when they are drinking, with responses: 1 to 2 units (coded as 0), 3 to 4 units (coded as 1), 5 to 6 units (coded as 2), 7 to 9 units (coded as 3) and 10+ units (coded as 4). Summing responses from both questions resulted in an alcohol use score ranging from 0 to 8. A score of ≥6 was considered to indicate higher risk alcohol consumption. A missing response was included in the lower risk/no alcohol consumption category (score of <6).

### Internalised homophobia

At month 12 and 24, participants were asked eight questions surrounding attitudes towards gay sexuality, based on the 26-item Internalised Homophobia Scale (four scales with internal reliabilities of 0.85, 0.69, 0.64 and 0.62).[22] Each question had the following response options: strongly agree, agree, neutral or uncertain, disagree and strongly disagree. The eight questions included the following: (i) 'I feel comfortable in gay bars', (ii) 'I feel comfortable being seen in public with an obviously transgender/gay person', (iii) 'I feel comfortable discussing homosexuality in a public situation', (iv) 'I feel comfortable being a transgender/gay man', (v) 'Homosexuality is morally acceptable to me', (vi) 'Even if I could change my sexual orientation I wouldn't', (vii) 'Obviously

effeminate homosexual men make me feel uncomfortable' and (viii) 'Social situations with transgender/gay men make me feel uncomfortable'. Responding with 'strongly agree' or 'agree' to questions (vii) or (viii), or with 'strongly disagree' or 'disagree' to questions (i), (ii), (iii), (iv), (v) or (vi) was considered to indicate internalised homophobia.

### Concealment of sexual identity

At month 12 and 24, participants were also asked how many of their (i) work colleagues, (ii) friends and (iii) close family, know that they are gay/transgender/have sex with men. Response options for groups (i) to (iii) included: all/almost all, >half, <half, few and none. Participants who did not report being 'out' to all/almost all friends, work colleagues and close family were considered to indicate concealment of their sexual identity.

### Statistical analysis

Prevalence of depressive symptoms (PHQ-9 ≥10) was compared between trial arms at each time-point using $\chi^2$ tests and across time-points (trial arms combined) using McNemar's $\chi^2$ test to account for paired responses.

Unadjusted and adjusted associations of demographic, socioeconomic and psychosocial (drinking and drug use) factors with depressive symptoms (PHQ-9 ≥10) at baseline were assessed using modified Poisson regression with a robust variance estimator, in order to produce prevalence ratios (PRs).[23] Seven socio-demographic factors assumed not to be on hypothesised causal pathways were adjusted for: Age group (<25, 25 to 29, 30 to 39 or 40+ years), born in the UK (yes or no/missing), sexual identity (gay or bisexual/straight), university education (yes or no/missing), ongoing relationship (yes or no/missing), London study clinic site (yes or no) and calendar time (year and calendar quarter). For each factor investigated, separate socio-demographic adjusted models were constructed.

Generalised estimating equations (GEEs) were used to assess the cross-sectional association of psychosocial factors (sexualised drug use, age at anal sex debut, IPV, internalised homophobia, 'outness') with depressive symptoms, including data from the 12 and 24 month questionnaires. GEEs were used to account for multiple responses from individual participants, and were fitted using Poisson models with a log link and compound symmetry for the correlational structure. In the GEEs, adjustment was made for the same set of baseline socio-demographic factors described above; with the exceptions that age and calendar time were included as time-updated variables, baseline 'ongoing relationship' was not adjusted for as this status may change over time and additional adjustment was made for follow-up time-point (month 12, month 24). Use of GEE models in this way represents a pooled cross-sectional analysis. A longitudinal analysis was not feasible as the number of 'incident' cases of PHQ-9 depression was low; of the 233 men who appeared to have no history of depression based on

PROUD data (PHQ-9 <10 at baseline and month 12 and no antidepressant use, as captured at baseline only), 22 reported clinically significant symptoms (PHQ-9 ≥10) at month 24.

For each measure and at each time-point, the proportion of missing responses was <5%. For psychosocial measures (depression, drinking, drug use, sexualised drug use, internalised homophobia and IPV), missing responses were considered to indicate that the event did not occur. There appeared to be a common response pattern in which only those experiences that had occurred were ticked on the questionnaire. Given the very low proportion of missing responses, missing values were considered to indicate a lower level of educational attainment (ie, no university degree), no ongoing relationship and no current employment. A sensitivity analysis was undertaken excluding missing values when defining each variable. The findings were very similar to the main analysis. All analyses were performed in Stata statistical software (V.13)[24] and reported according to the Strengthening the Reporting of Observational Studies in Epidemiology guidelines.

## RESULTS

There were 544 participants enrolled in PROUD. In total, 540 completed a baseline questionnaire, of whom 410 (76%) completed a 12 month questionnaire, and 333 (62%) completed a 24 month questionnaire. Overall, 96% of men identified as gay and 3% as bisexual (includes three trans women). Eighty-two per cent of men were of white ethnicity; median age was 35 years (IQR: 29 to 42 years) at baseline (table 1). The prevalence of psychosocial measures collected at baseline and follow-up time-points are presented in tables 1 and 2. Briefly, lifetime IPV was reported by 45% of participants for victimisation and 20% for perpetration at month 12, and 42% of participants indicated internalised homophobia at month 12. The prevalence of CAS with greater-than or equal to two partners in the past 3 months was: 76% at baseline, 74% at month 12 and 75% at month 24.

### Prevalence of depressive symptoms across trial arms and time-points

The prevalence of depressive symptoms was 9.1% (49/540) at baseline, 14.4% (59/410) at month 12 and 14.4% (48/333) at month 24. In terms of changes in individual depression status between time-points, among the 410 men who completed the baseline and 12 month questionnaire, 38 men reported depressive symptoms (PHQ-9 ≥10) at baseline, 22 (57.9%) of whom continued to report this measure of depression at month 12. Of the 372 men who did not report depressive symptoms at baseline, 37 (10.0%) reported it at month 12. Among the 333 men who completed the baseline and 24 month questionnaire, 32 men reported depressive symptoms (PHQ-9 ≥10) at baseline, 11 (34.4%) of whom continued to report this measure of depression at month 24. Of the

**Table 1** Prevalence of demographic, socioeconomic and psychosocial (drinking and drug use) factors at baseline in PROUD

| Measure | Category | Baseline ((N=540) n (%)) |
|---|---|---|
| Calendar time (year & calendar quarter((Q)) of baseline data collection | 2012 Q4 | 8 (1.5%) |
| | 2013 Q1 | 63 (11.7%) |
| | 2013 Q2 | 85 (15.7%) |
| | 2013 Q3 | 113 (20.9%) |
| | 2013 Q4 | 119 (22.0%) |
| | 2014 Q1 | 125 (23.2%) |
| | 2014 Q2 | 27 (5.0%) |
| Age | <25 | 54 (10.0%) |
| | 25-29 | 96 (17.8%) |
| | 30-34 | 114 (21.1%) |
| | 35-39 | 97 (18.0%) |
| | 40-44 | 81 (15.0%) |
| | 45+ | 98 (18.2%) |
| Born in the UK and ethnicity | Yes, white | 287 (53.4%) |
| | Yes, BAME | 35 (6.5%) |
| | No, white | 152 (28.3%) |
| | No, BAME | 64 (11.9%) |
| Self-reported sexual identity | Gay | 513 (95.7%) |
| | Bisexual | 17 (3.2%) |
| | Straight | 6 (1.1%) |
| University education | Yes | 327 (60.6%) |
| | No/missing | 213 (39.4%) |
| Employed | Yes | 439 (81.3%) |
| | No/missing | 101 (18.7%) |
| Ongoing relationship | Yes | 246 (45.6%) |
| | No/missing | 294 (54.4%) |
| Study region | London | 375 (69.4%) |
| | Outside London | 165 (30.6%) |
| Higher-risk drinking | No/missing | 367 (68.0%) |
| | Yes | 173 (32.0%) |
| Recreational drug use (past 3 months) | 0/missing | 148 (27.4%) |
| | 1 | 87 (16.1%) |
| | 2-4 | 159 (29.4%) |
| | 5+ | 146 (27.0%) |
| Chemsex-associated drug use (past 3 months) | No/missing | 309 (57.2%) |
| | Yes | 231 (42.8%) |
| Crystal meth (past 3 months) | No/missing | 442 (81.9%) |
| | Yes | 98 (18.2%) |
| Mephedrone (past 3 months) | No/missing | 343 (63.5%) |
| | Yes | 197 (36.5%) |
| GHB/GBL (past 3 months) | No/missing | 371 (68.7%) |
| | Yes | 169 (31.3%) |

Continued

**Table 1** Continued

| Measure | Category | Baseline ((N=540) n (%)) |
|---|---|---|
| Ketamine (past 3 months) | No/missing | 451 (83.5%) |
| | Yes | 89 (16.5%) |
| Cocaine (past 3 months) | No/missing | 401 (74.3%) |
| | Yes | 139 (25.7%) |
| Ecstasy/MDMA (past 3 months) | No/missing | 450 (83.3%) |
| | Yes | 90 (16.7%) |

BAME, Black, Asian and minority ethnic; GHB/GBL, gamma-hydroxybutyrate/gamma-butyrolactone; MDMA, 3,4-methylenedioxymethamphetamine.

301 men who did not report depressive symptoms at baseline, 37 (12.3%) reported it at month 24. Finally, among the 307 men who completed both the 12 and 24 month questionnaire, of 43 men who reported depressive symptoms (PHQ-9 ≥10) at month 12, 16 (37.2%) continued to report this measure of depression at month 24. Of the 264 men who did not report depression at month 12, 29 (11.0%) reported it at month 24.

The prevalence of depressive symptoms was slightly higher in the immediate than the deferred PrEP group at baseline (10.6% vs 7.5%) and at subsequent time-points (16.7% vs 11.9% at month 12; 16.5% vs 11.9% at month 24). Differences between randomised groups were not statistically significant at any time-point (p≥0.2). The within-person increase in prevalence from 9.3% at baseline to 14.4% at month 12 (among 410 men) and from 9.6% at baseline to 14.4% at month 24 (among 333 men) was significant (p≤0.04), but there was no difference from 14.0% at month 12 to 14.7% at month 24 (among 307 men).

Overall, 8.5% (46/540) of men reported antidepressant use at baseline; 20.4% (n=10) of men with depressive symptoms (PHQ-9 ≥10) and 7.3% (n=36) of men without. There were 85 men (15.7%) who either had depressive symptoms (PHQ-9 ≥10) or reported antidepressant use, of whom 54.1% (n=46) were receiving antidepressant medication.

### Association of socioeconomic and psychosocial (drinking and drug use) factors with depressive symptoms (PHQ-9 ≥10) at baseline

Younger age, unemployment, methamphetamine use in the past 3 months and completion of baseline questionnaire before the first quarter of 2013 was associated with higher prevalence of depressive symptoms in unadjusted and adjusted analysis (see tables 3 and 4).

### Association of psychosocial factors (sexualised drug use, age at sex debut, IPV, internalised homophobia, 'outness') with depressive symptoms at the 12 and 24 month questionnaire

The prevalence of depressive symptoms was two to three-fold higher among men who reported: ever having been

**Table 2** Prevalence of psychosocial factors (sexualised drug use, age at sex debut, IPV, internalised homophobia, 'outness') at follow-up time-points in PROUD

| Measure | Category | 12 month (N=410) n (%) | 24 month (N=333) n (%) |
|---|---|---|---|
| Had sex after using recreational drugs (past 3 months) | No/missing | 186 (45.4%) | 174 (52.3%) |
| | Yes | 224 (54.6%) | 159 (47.8%) |
| Age ≤15 years at anal sex debut | No | 307 (77.7%) | 251 (78.2%) |
| | Yes | 88 (22.3%) | 70 (21.8%) |
| Age <13 years at anal sex debut | No | 373 (94.4%) | 302 (94.1%) |
| | Yes | 22 (5.6%) | 19 (5.9%) |
| Any IPV victimisation | No/missing | 226 (55.1%) | 199 (59.8%) |
| | Yes | 184 (44.9%) | 134 (40.2%) |
| Any IPV victimisation in the past year | No/missing | 346 (84.4%) | 284 (85.3%) |
| | Yes | 64 (15.6%) | 49 (14.7%) |
| Any IPV perpetration | No/missing | 330 (80.5%) | 273 (82.0%) |
| | Yes | 80 (19.5%) | 60 (18.0%) |
| Any IPV perpetration in the past year | No/missing | 378 (92.2%) | 310 (93.1%) |
| | Yes | 32 (7.8%) | 23 (6.9%) |
| Internalised homophobia | No/missing | 239 (58.3%) | 195 (58.6%) |
| | Yes | 171 (41.7%) | 138 (41.4%) |
| Concealment of sexual identity | No | 206 (51.6%) | 166 (50.6%) |
| | Yes | 193 (48.4%) | 162 (49.4%) |

IPV, intimate partner violence.

a victim of IPV, IPV victimisation in the past year, ever having perpetrated IPV, perpetration of IPV in the past year, internalised homophobia and concealment of sexual identity, in unadjusted and adjusted analysis (see table 5).

### Exploration of changes in depression prevalence

The baseline prevalence of suicidal ideation on PHQ-9 was significantly higher for men who completed their questionnaire before 1 April 2013 (n=71) compared with men whose completion was on or after 1 April 2013, from which time they were asked to disclose their response (n=469) (19.7% vs 9.6%; Pearson $\chi^2$ test p=0.011). Similarly, the baseline prevalence of depressive symptoms (PHQ-9 ≥10) was significantly higher among men who completed their questionnaire before versus on/after 1 April 2013 (21.1% vs 7.3%; Pearson $\chi^2$ test p<0.001). A possible calendar time effect can also be seen in table 3, and this relationship remained after adjusting for sociodemographics. There was evidence of an increase in depression from baseline to month 12 for the 353 men with baseline completion on/after 1 April 2013 and 12 month data (7.9% vs 14.7%), but not for the 57 men with baseline completion before 1 April 2013 and 12 month data (17.5% vs 12.3%; McNemar's $\chi^2$ test p values 0.366 and <0.001, respectively). When investigating the modifying effect of baseline completion date on the association between follow-up time-point and depressive symptoms in GEEs, the interaction p value was 0.004: for month 12 versus baseline, PR 0.61 95% CI: 0.28 to 1.34 for the time period before 1 April 2013 and PR 1.97 95% CI:

1.41 to 2.75 for the time period on/after 1 April 2013, respectively.

### DISCUSSION

In the PROUD trial of GBMSM reporting very high levels of CAS partners, the prevalence of depressive symptoms by PHQ-9 ranged from 9.1% at baseline to 14.4% at months 12 and 24. Younger men, those who were not employed, those who reported methamphetamine use, IPV, internalised homophobia and concealment of sexual identity, were more likely to report depressive symptoms. Depression prevalence did not differ significantly between trial arms at any time-point, which was in line with that reported in the iPrEX (placebo controlled) trial of PrEP.[25]

The link between psychosocial factors, including depressive symptoms, and sexual risk behaviour among GBMSM in PROUD has been addressed in a separate paper.[26] Unlike in previous studies,[16 17] no link was found between depression and CAS. Given the PROUD inclusion criteria, all participants reported recent CAS partners. It may be that symptoms of depression do not distinguish between men reporting higher levels of CAS partners and lower levels of CAS partners.

In the current study, findings suggest that some men may have underreported depressive symptoms at baseline after April 2013. It is possible that newly enrolled participants may have been worried about their inclusion in

**Table 3** Unadjusted and adjusted associations of demographic and socioeconomic factors with depressive symptoms at baseline in PROUD

| N=540 men at baseline | | Depressive symptoms (PHQ-9 ≥10) | | | | | |
|---|---|---|---|---|---|---|---|
| | | % | P value* | Unadjusted PR (95% CI) | Overall p value† | Adjusted PR (95% CI)‡ | Overall p value† |
| Calendar time (year & calendar quarter (Q))§ | 2012 Q4 – 2013 Q1 | 21.1% | 0.001 | 1 | 0.003 | 1 | 0.004 |
| | 2013 Q2 | 2.4% | 0.20¶ | 0.11 (0.03 to 0.47) | 0.25¶ | 0.12 (0.03 to 0.47) | 0.26¶ |
| | 2013 Q3 | 7.1% | | 0.34 (0.15 to 0.75) | | 0.34 (0.15 to 0.76) | |
| | 2013 Q4 | 8.4% | | 0.40 (1.19 to 0.84) | | 0.39 (0.18 to 0.83) | |
| | 2014 Q1 – Q2 | 9.2% | | 0.44 (0.22 to 0.85) | | 0.45 (0.23 to 0.88) | |
| Age | <25 | 22.2% | 0.003 | 3.63 (1.44 to 9.13) | 0.006 | 3.49 (1.44 to 8.44) | 0.003 |
| | 25-29 | 8.3% | 0.002¶ | 1.36 (0.49 to 3.78) | 0.003¶ | 1.22 (0.43 to 3.51) | 0.004¶ |
| | 30-34 | 12.3% | | 2.01 (0.80 to 5.02) | | 1.97 (0.78 to 4.97) | |
| | 35-39 | 6.2% | | 1.01 (0.34 to 3.03) | | 0.96 (0.31 to 3.00) | |
| | 40-44 | 3.7% | | 0.60 (0.16 to 2.35) | | 0.60 (0.16 to 2.31) | |
| | 45+ | 6.1% | | 1 | | 1 | |
| Born in the UK and white ethnicity | Yes, white | 9.1% | 0.32** | 1 | 0.37 | 1 | 0.44 |
| | Yes, BAME | 14.3% | | 1.58 (0.65 to 3.84) | | 0.93 (0.41 to 2.09) | |
| | No, white | 6.6% | | 0.73 (0.36 to 1.47) | | 0.77 (0.36 to 1.65) | |
| | No, BAME | 12.5% | | 1.38 (0.65 to 2.91) | | 1.59 (0.72 to 3.54) | |
| Self-reported sexual identity†† | Gay | 9.0% | 0.46** | 1 | 0.5 | 1 | 0.64 |
| | Bisexual/Straight | 13.0% | | 1.45 (0.49 to 4.33) | | 1.26 (0.48 to 3.31) | |
| University education | Yes | 8.9% | 0.84 | 1 | 0.84 | 1 | 0.94 |
| | No/missing | 9.4% | | 1.06 (0.61 to 1.82) | | 0.98 (0.54 to 1.77) | |
| Employed | Yes | 6.8% | <0.001 | 1 | <0.001 | 1 | 0.001 |
| | No/missing | 18.8% | | 2.75 (1.62 to 4.69) | | 2.52 (1.45 to 4.37) | |
| Ongoing relationship | Yes | 6.5% | 0.06 | 1 | 0.06 | 1 | 0.15 |
| | No/missing | 11.2% | | 1.73 (0.97 to 3.06) | | 1.50 (0.86 to 2.63) | |
| Study region | London | 9.3% | 0.75 | 1 | 0.75 | 1 | 0.66 |
| | Outside London | 8.5% | | 0.91 (0.50 to 1.64) | | 0.86 (0.44 to 1.69) | |

*Pearson $\chi^2$ test.
†P value by Wald test using modified Poisson regression with a robust variance estimator in order to produce adjusted prevalence ratios.
‡Age (included as four categories:<25, 25 to 29, 30 to 39, 40+), born in the UK, sexual identity (gay or bisexual/straight), university education, relationship status, London study clinic site and calendar time.
§Some calendar quarters (Q4 2012 & Q1 2013; Q1-Q2 2014) were combined due to small cell counts.
¶Test for trend.
**Fisher's exact test.
††A dichotomized version of self-reported sexual identity is investigated in analysis due to very small numbers: gay or bisexual/straight.
BAME, Black, Asian and minority ethnic; PHQ-9, Patient Health Questionnaire-9; PR, prevalence ratio.

the trial if they disclosed thoughts of suicide or self-harm to the investigator, and that such concerns may have dissipated over time once participation in the study was established.

Patient and public involvement activities with a small group of PROUD participants (n=13), in the form of a short qualitative survey, provided some participant perspectives on the change in depressive symptoms over time. Some highlighted the possibility that greater familiarity with the questionnaire and study staff over time may have resulted in increased reporting of depressive symptoms at follow-up visits. Additionally, partaking in a clinical trial was thought to be one of the processes that helps individuals recognise/acknowledge their own depression in an environment where access to support is provided. Other suggested explanations for an actual increase in depressive symptoms included the possibility of greater socioeconomic hardship in a period of national financial instability, risk taking behaviours in relation to CAS and chemsex or PrEP-related stigma. None of the 13 participants specifically theorised that engagement in chemsex increased over the study, but chemsex was a common feature in discourse on depression. Although these participants were informed of the lack of difference

**Table 4** Unadjusted and adjusted associations of psychosocial factors (drinking and drug use) with depressive symptoms at baseline in PROUD

| n=540 men at baseline | | Depressive symptoms (PHQ-9 ≥10) | | | | | |
|---|---|---|---|---|---|---|---|
| | | % | P value* | Unadjusted PR (95% CI) | Overall p value† | Adjusted‡ PR (95% CI) | Overall p value† |
| Higher-risk drinking | No/missing | 8.5% | 0.46 | 1 | | 1 | 0.48 |
| | Yes | 10.4% | | 1.23 (0.71 to 2.14) | 0.46 | 1.25 (0.68 to 2.29) | |
| Recreational drug use (past 3 months) | 0/missing | 10.1% | 0.11 | 1 | 0.13 | 1 | 0.32 |
| | 1 | 4.6% | 0.40§ | 0.45 (0.16 to 1.32) | 0.45§ | 0.57 (0.20 to 1.64) | 0.32§ |
| | 2–4 | 6.9% | | 0.68 (0.32 to 1.44) | | 0.89 (0.39 to 2.02) | |
| | 5+ | 13.0% | | 1.28 (0.68 to 2.43) | | 1.34 (0.71 to 2.56) | |
| Chemsex-associated drug use | No/missing | 8.1% | 0.36 | 1 | 0.36 | 1 | 0.33 |
| | Yes | 10.4% | | 1.28 (0.75 to 2.19) | | 1.31 (0.77 to 2.23) | |
| Crystal meth (past 3 months) | No/missing | 7.7% | 0.02 | 1 | 0.02 | 1 | 0.03 |
| | Yes | 15.3% | | 1.99 (1.13 to 3.51) | | 1.95 (1.06 to 3.61) | |
| Mephedrone (past 3 months) | No/missing | 9.0% | 0.97 | 1 | 0.97 | 1 | 0.96 |
| | Yes | 9.1% | | 1.01 (0.58 to 1.76) | | 0.99 (0.56 to 1.73) | |
| GHB/GBL (past 3 months) | No/missing | 8.4% | 0.39 | 1 | 0.39 | 1 | 0.46 |
| | Yes | 10.7% | | 1.27 (0.73 to 2.21) | | 1.23 (0.72 to 2.11) | |
| Ketamine (past 3 months) | No/missing | 8.4% | 0.24 | 1 | 0.24 | 1 | 0.27 |
| | Yes | 12.4% | | 1.47 (0.78 to 2.76) | | 1.43 (0.76 to 2.71) | |
| Cocaine (past 3 months) | No/missing | 8.5% | 0.41 | 1 | 0.41 | 1 | 0.64 |
| | Yes | 10.8% | | 1.27 (0.72 to 2.27) | | 1.14 (0.66 to 1.97) | |
| Ecstasy/MDMA (past 3 months) | No/missing | 8.4% | 0.26 | 1 | 0.25 | 10 | 0.41 |
| | Yes | 12.2% | | 1.45 (0.77 to 2.72) | | 1.31 (0.69 to 2.49) | |

*Pearson $\chi^2$ test.
†P value by Wald test using modified Poisson regression with a robust variance estimator in order to produce adjusted prevalence ratios.
‡Age (included as four categories:<25, 25 to 29, 30 to 39, 40+), born in the UK, sexual identity (gay or bisexual/straight), university education, relationship status, London study clinic site and calendar time.
§Test for trend.
¶Fisher's exact test.
GHB/GBL, gamma-hydroxybutyrate/gamma-butyrolactone; MDMA, 3,4-methylenedioxymethamphetamine; PHQ-9, Patient Health Questionnaire-9; PR, prevalence ratio.

**Table 5** Unadjusted and adjusted associations of psychosocial factors (sexualised drug use, age at sex debut, IPV, internalised homophobia, 'outness') with depressive symptoms using 12 and 24 month PROUD questionnaire data in GEE models

| n=436 men, observations=743 (using data from men who completed the 12 month or 24 month questionnaire in GEE models) | | Depressive symptoms (PHQ-9 ≥10) | | | |
|---|---|---|---|---|---|
| | | Unadjusted PR (95% CI) | Overall p value* | Adjusted† PR (95% CI) | Overall p value* |
| Had sex after using recreational drugs (past 3 months) N=436‡, Obs=743§ ; N=432¶, Obs=737** | No/missing | 1 | | 1 | |
| | Yes | 1.38 (0.93 to 2.06) | 0.11 | 1.38 (0.92 to 2.07) | 0.12 |
| Age ≤15 years at anal sex debut N=433‡, Obs=716§ ; N=429¶, Obs=710** | No | 1 | | 1 | |
| | Yes | 1.27 (0.80 to 2.02) | 0.31 | 1.20 (0.75 to 1.94) | 0.45 |
| Age <13 years at anal sex debut N=433‡, Obs=716§ ; N=429¶, Obs=710** | No | 1 | | 1 | |
| | Yes | 0.71 (0.26 to 1.94) | 0.51 | 0.72 (0.27 to 1.97) | 0.526 |
| Any IPV victimisation N=436‡, Obs=743§ ; N=432¶, Obs=737** | No/missing | 1 | | 1 | |
| | Yes | 2.45 (1.63 to 3.67) | <0.001 | 2.56 (1.70 to 3.85) | <0.001 |
| Any IPV victimisation in the past year N=436‡, Obs=743§ ; N=432¶, Obs=737** | No/missing | 1 | | 1 | |
| | Yes | 2.82 (1.88 to 4.22) | <0.001 | 2.96 (1.97 to 4.46) | <0.001 |
| Any IPV perpetration N=436‡, Obs=743§ ; N=432¶, Obs=737** | No/missing | 1 | | 1 | |
| | Yes | 2.83 (1.89 to 4.22) | <0.001 | 2.86 (1.90 to 4.30) | <0.001 |
| Any IPV perpetration in the past year N=436‡, Obs=743§ ; N=432¶, Obs=737** | No/missing | 1 | | 1 | |
| | Yes | 3.40 (2.13 to 5.41) | <0.001 | 3.36 (2.06 to 5.47) | <0.001 |
| Internalised homophobia N=436‡, Obs=743§ ; N=432¶, Obs=737** | No/missing | 1 | | 1 | |
| | Yes | 1.92 (1.30 to 2.83) | 0.001 | 1.91 (1.28 to 2.83) | 0.001 |
| Concealment of sexual identity N=434‡, Obs=727§ ; N=431¶, Obs=722** | No | 1 | | 1 | |
| | Yes | 1.75 (1.17 to 2.62) | 0.007 | 1.71 (1.13 to 2.59) | 0.012 |

*P value by Wald test using GEE models.

†Age (included as four categories:<25, 25 to 29, 30 to 39, 40+), born in the UK, sexual identity (gay or bisexual/straight), university education, London study clinic site, calendar time (year & calendar quarter) and follow-up time-point (month 12, month 24).

‡Number of men contributing observations to the unadjusted model.

§Number of observations examined in the unadjusted model.

¶Number of men contributing observations to the adjusted model.

**Number of observations examined in the adjusted model.

GEE, generalised estimating equations; IPV, intimate partner violence; Obs, observations; PHQ-9, Patient Health Questionnaire-9; PR, prevalence ratio.

between the immediate and deferred PrEP groups with regards to depression prevalence, some men cited that PrEP related side-effects and stigma, as well as fear of loss of access to PrEP after study completion, may have increased over time.

The depression prevalence observed in PROUD at month 12 (14.4%) and 24 (14.4%) is comparable to that found in the Attitudes to and Understanding of Risk of Acquisition of HIV (AURAH) cross-sectional survey of HIV-negative GBMSM attending GUM services in England, 2013 to 2014 (PHQ-9≥10): 12.4% (n=166/1340) and 15.0% (n=122/815) among GBMSM reporting recent sex (with or without a condom) and recent CAS respectively.[17] Depression prevalence at annual follow-up visits in PROUD was also broadly similar to other studies of GBMSM in the UK,[13–18] although comparison is made difficult due to different study settings and measures of depression, and the fact that people with HIV were included in most studies (table 6). The prevalence of depressive symptoms observed at follow-up in PROUD was almost twice that observed in a random sample of men and women in England (2007/2008); 7% reported PHQ-9 ≥10,[27] although socio-demographic differences may confound this comparison.

Although most studies based prevalence estimates on depressive symptoms only, the proportion of participants with evidence of current depression is higher when taking into account treatment for depression as well as symptoms. In the PROUD trial, 36 (78%) of the 46 men reporting antidepressant use at baseline were not positive for PHQ-9 depressive symptoms, and the prevalence of depression when considering symptoms or treatment was 15.7%.

In recent European studies, younger age, markers of lower socioeconomic status, lower levels of a supportive network, reporting a bisexual identity[14 17 28] and recreational drug use,[17 28] appear to be important factors for depression in the GBMSM literature. Mephedrone and

**Table 6** Prevalence of depression on symptom questionnaires in studies of UK GBMSM

| Study | Recruitment dates | Sampling strategy | Size of GBMSM sample | Measure of depression | Depression prevalence |
|---|---|---|---|---|---|
| Cross-sectional survey of GBMSM in London[16] | 1999 | Consecutive sampling of GBMSM attending a GUM clinic | 122 | HADS score of ≥8 | 18.8% |
| Cross-sectional survey of GBMSM in London[18] | 2010–2011 | Consecutive sampling of GBMSM attending GUM clinics | 519 | HADS score of ≥8 | 12% |
| Natsal-3[13] | 2010–2012 | Multistage probability sampling of the general British population | 190 | PHQ-2 score of ≥3 | 8.9% |
| Cross-sectional survey of GBMSM in London[15] | 2016 | Volunteer sampling of users of a geosocial-networking app for GBMSM | 179 | PHQ-2 score of ≥3 | 22.3% |
| Gay & Bisexual Men's Health Survey[14] | 2011 | Snowball sampling of GBMSM | 5416 | PHQ-9 score of ≥10 | 21.3% |
| AURAH[17] | 2013–2014 | Consecutive sampling of HIV-negative GBMSM attending GUM clinics in England | 1340 sexually active GBMSM | PHQ-9 score of ≥10 | 12.4% |
| PROUD | 2012–2014 baseline | Volunteer sampling of HIV-negative GBMSM attending GUM clinics and community organisations, and GBMSM users of social media | 540 | PHQ-9 score of ≥10 | 9.1% |
| | 2013–2015 month 12 | | 410 | | 14.4% |
| | 2014–2016 month 24 | | 333 | | 14.4% |

app, application; GBMSM, gay, bisexual and other men who have sex with men; GUM, genitourinary medicine; HADS, Hospital Anxiety and Depression Scale; Natsal-3, Third National Survey of Sexual Attitudes and Lifestyles; PHQ, Patient Health Questionnaire.

crystal methamphetamine use have also been linked to acute symptoms of depression in a qualitative study of 30 gay men in London.[9] Some men reported a sense of loss of intimacy with a sexual partner when engaging in chemsex. There is also some evidence to suggest that group sex environments may leave some individuals vulnerable to mistreatment particularly if drugs are used, given their impact on inhibition and self-regulation.[29] Chemsex associated drug use has been found to be strongly associated with depressive symptoms among UK GBMSM in the AURAH study, including after adjustment for socio-demographic factors.[17] Although the pattern of associations was similar in the PROUD trial, some of the differences observed (ie, between sexual identities, number of recreational drugs used and any chemsex drug use) were not statistically significant. It may be that due to the possible underreporting of depression at baseline and the limited sample size of this trial, differences that may exist were not detected. It is also important to note that this sample of men is unique — they all reported very high levels of CAS partners and agreed to participate in a randomised control trial. Therefore, associations may differ from other samples of GBMSM.

There is some evidence to suggest that CSA is associated with depression among GBMSM in the USA.[30] In the PROUD trial, a very young age at anal sex debut was not associated with depression. This measure may be a poor proxy for CSA as it is not likely to reflect or pick up all sexual abuse experienced. Strong associations were found of any IPV victimisation and any IPV perpetration with depressive symptoms in the PROUD trial. Evidence from a recent meta-analysis of mostly US studies

suggests that exposure to any kind of IPV victimisation is associated with increased odds of depressive symptoms (pooled OR 1.52, 95% CI: 1.24 to 1.86).[8] Similar findings were observed in six recent studies.[5 18 31–34] In one UK study (GUM clinic attendees, 2010 to 2011), depression (Hospital Anxiety and Depression Scale ≥8) was significantly elevated among men reporting IPV perpetration in the past year (20.7% vs 11.5%), but the association was attenuated after adjusting for socioeconomic factors (OR 3.7; 95% CI: 1.0 to 14.6; p=*0.060*).[18] Adjusting for income may in part have contributed to the attenuation. Financial insecurity may be closely linked to depression, and financial inequalities within intimate partnerships may increase the risk of controlling and abusive behaviours. It was not possible to investigate the confounding effect of income/financial security in PROUD.

As described by syndemic theory, recreational drug use, IPV and depression may influence each other in a cyclical pattern, with implications for an exaggerated risk of poor health outcomes.[5] IPV may distort one's perceived self-worth and capacity to influence important outcomes in one's life, which may lead to depressive symptoms,[35] in turn depression may heighten one's vulnerability to dysfunctional relationship dynamics as adaptive coping mechanisms are distorted.[36–38] Recreational drug use may cause symptoms of depression, which in turn results in further use of drugs to induce a state of cognitive release and escape from symptoms.

In line with minority stress theory, a measure of internalised homophobia and concealment of sexual identity was strongly associated with depressive symptoms in the PROUD trial. Due to the significance of early life

socialisation experiences, the minority stress model suggests that residual internalised homophobia may be a permanent fixture throughout the life-course for a sexual minority individual. If individuals integrate a high degree of antigay attitudes into their perception of self, depression may follow.[2] This may in part be the result of constant preoccupation with, and cognitive effort involved in, hiding one's identity,[39] which is a profoundly heavy burden to carry. Three factors are hypothesised to reverse the process of internalised homophobia and reduce the risk of mental health distress: (i) family support, (ii) personal resources including personality characteristics and (iii) group resources/social structural factors including community solidarity and cohesiveness. Very few studies have collected this information. There is a need to better understand correlates of these protective factors, as this may be useful for informing interventions to reduce the burden of depression among sexual minority men.

### Strengths and limitations

This is the first UK study to collect data on depression prevalence over time among GBMSM, and investigate associations with a range of psychosocial factors. A change in methodology part way through baseline data collection may have affected the reporting of depressive symptoms, limiting the validity of findings at baseline. Given the relatively small sample size of the PROUD trial, the examination of factors associated with depression was based on cross-sectional analysis, utilising all available observations. Some participants were not included in the GEE analysis of psychosocial correlates due to missing questionnaires at months 12 and 24. However, men with and without depressive symptoms (PHQ-9 ≥10) at baseline had similar proportions completing the 12 month questionnaire (22.5% vs 24.2%; $\chi^2$ test p=0.780) and those with and without depressive symptoms at the 12 month questionnaire had similar proportions completing the 24 month questionnaire (27.1% vs 24.8%; $\chi^2$ test p=0.702). When stratifying by trial arm, there remained no difference between men with and without depressive symptoms in terms of proportion completing follow-up questionnaires. Even after including repeated observations in GEE models, the analysis may have lacked power to accurately detect the presence of some associations. Furthermore, the direction of any causality may be unclear for some associations. For this reason, as well as the possibility of unmeasured confounders, inferences regarding causal factors for depression are limited. Although chemsex drug use was investigated in this study, the cultural phenomenon of chemsex was not explicitly measured, as participants were not asked whether these drugs were taken to enhance sexual experiences.

There have been no longitudinal studies that are designed and powered to investigate depression among GBMSM. Future longitudinal analyses would be useful in investigating the suggested causal pathways in this paper.

Such a study would require careful consideration in how to measure incident depression, as new symptoms may indicate first episode or recurrent depression. Therefore, capturing data on previous diagnoses and treatment is important. Following participants from a young age is likely to yield the most accurate data with regards to depression incidence. Finally, there is a need to examine associations with measures of family and community support, income and abuse in childhood, as well as the role of these factors in relationships investigated in this paper.

### CONCLUSIONS AND IMPLICATIONS

In the context of interviewer administered surveys or surveys in which interviewers/study staff are made aware of individual responses, participants may be reluctant to disclose information on depressive symptomatology. This may be particularly true in clinical trial settings that are subject to strict inclusion/exclusion criteria, as eligible participants may perceive the disclosure of current depressive symptoms as prohibiting study inclusion. Researchers should be aware of the potential limitations to reporting depression prevalence under these circumstances.

Our findings add to the growing body of research in high-income countries suggesting a high concomitant burden of sexual minority related stress, substance use and IPV with depression among GBMSM. Since these psychosocial measures are relevant to GBMSM at high-risk of HIV and seeking PrEP, it is recommended that training on awareness and enquiry about psychosocial issues should be enhanced in sexual health services. This should include referral to substance use services and culturally appropriate structures set up to address IPV among same-sex male couples. Furthermore, findings suggest that new interventions addressing sexual minority related stresses are needed in order to foster greater community affiliation and support, and curtail stress. Intervening with sexual minority youths in schools and/or in lesbian, gay, bisexual and transgender+youth groups may prevent the internalising of homophobic attitudes and the onset of mental health problems. Although it is important to be cognizant of the unique forms of stress experienced by sexual minority men, findings from PROUD also emphasise the link between socioeconomic disadvantage and poor mental health. Therefore, emphasis on alleviating economic hardship together with efforts to promote self-acceptance and pride towards one's sexual orientation is recommended when addressing depression among sexual minority men.

In many countries, sexual minorities are being accepted and celebrated more so than ever in history.[40–43] This social climate provides a solid platform from which to launch extensive efforts to eliminate homophobic discrimination in society at large. Such a structural change is likely to be profoundly beneficial to the health and well-being of sexual minority people.

Efforts may include national educational campaigns and specialised training for healthcare providers on homophobia and heterosexism.

**Author affiliations**
[1]Institute for Global Health, University College London, London, UK
[2]MRC CTU, University College London, London, UK
[3]Chelsea and Westminster Healthcare NHS Trust, London, UK
[4]Homerton University Hospital NHS Trust, London, UK
[5]Brighton & Sussex University Hospital NHS Trust, Brighton & Sussex Medical School, Brighton, UK
[6]Department of Global Health and Development, London School of Hygiene and Tropical Medicine, London, UK

**Acknowledgements** We thank all PROUD trial participants. The PROUD study group: David I. Dolling, Monica Desai, Alan McOwan, Richard Gilson, Amanda Clarke, Martin Fisher, Gabriel Schembri, Ann K. Sullivan, Nicola Mackie, Iain Reeves, Mags Portman, Vanessa Apea, John Saunders, Julie Fox, Jake Bayley, Michael Brady, Killian Quinn, Christine Bowman, Clarie Dewsnap, Charles J. Lacey, Stephen Taylor, David White, Simone Antonucci, Mitzy Gafos, Sheena McCormack, Owen N. Gill, David T. Dunn and Anthony Nardone. PROUD clinic teams: Drew Clark, Paul Davis, James Hand, Machel Hunt, Rebecca Neale, Jackie O'Connell, Liat Sarner, John Saunders, Louise Terry, Angelina Twumasi, Salina Tsui, Dayan Vijeratnam, Ryan Whyte, Andy Williams, Sian Gately, Gerry Gilleran, Jill Lyons, Chris McCormack, Katy Moore, Cathy Stretton, Alex Acheampong, Michael Bramley, Marion Campbell, Ruby Chowdhry, Stewart Eastwood, Babs Fennell, Wendy Hadley, Kerry Hobbs, Sarah Kirk, Nicky Perry, Charlotte Rawlinson, Celia Richardson, Claire Richardson, Mark Roche, Emma Simpkin, Simon Shaw, Elisa Souto, Julia Williams, Elaney Youssef, Tristan Barber, Cindy Eliot, Serge Fedele, Chris Higgs, Kathryn McCormick, Alexandra Meijer, Sam Pepper, Jane Rowlands, Gurmit Singh, Alfredo SolerCarracedo, Sonali Sonecha, David Taylor, Lervina Thomas, Frederick Attakora, Marina Bourke, Richard Castles, Rebecca Clark, Anke De-Masi, Veronica Espa, Rumbidzai Hungwe, Martin Lincoln, Sifiso Mguni, Rhianon Nevin-Dolan, Hannah Alexander, Lucy Campbell, Sophie Candfield, Shema Doshi, Olivia Liddle, Larissa Mulka, Priyanka Saigal, James Stevenson, James Boateng, Brynn Chappell, Susanna Currie, Carolyn Davies, Dornubari Lebari, Matthew Phillips, Lisa Southon, Sarah Thorpe, Anna Vas, Chris Ward, Claire Warren, Stephanie Yau, Alejandro Arenas-Pinto, Asma Ashraf, Matthew Bolton, Lewis Haddow, Sara McNamara, Ana Milinkovic, June Minton, Dianne Morris, Clare Oakland, Steve O'Farrell, Pierre Pellegrino, Sarah Pett, Nina Vora, Carmel Young, Taris Zarko-Flynn, Wilbert Ayap, Ling Jun Chen, Adam Croucher, Sarah Fidler, Kristin Kuldanek, Ken Legg, Agathe Leon, Nadia Naous, Severine Rey, Judith Zhou, Margaret-Anne Bevan, Nina Francia, Eleanor Hamlyn, Lisa Hurley, Helen Iveson, Isabelle Jendrulek, Tammy Murray, Alice Sharp, Andrew Skingsley, Chi Kai Tam, Al Teague, Caroline Thomas, Juan-Manuel Tiraboschi, Christine Brewer, Richard Evans, Jan Gravely, Gary Lamont, Fabiola Martin, Georgina Morris, Sarah Russell-Sharpe, John Wightman, Anthony Bains, Gill Bell, Terry Cox, Charlie Hughes, Hannah Loftus, Naomi Sutton, Debbie Talbot, Vince Tucker.

**Contributors** Conceived and designed the study: MG, FCL, ARM, SM, DD, EW, AR, AP, LS, AC, AKS, IR. Analysed the data: ARM. Wrote the paper: ARM, MG, FCL. All authors read and approved the final manuscript.

**Funding** Sheena McCormack and David Dunn were supported by a Medical Research Council grant (MRC_UU_12023/23).

**Competing interests** The PROUD study was provided drug free of charge by Gilead Sciences plc which also distributed it to participating clinics and provided funds for additional diagnostic tests for HCV and drug levels. AP has received payments for presentations made at meetings sponsored by Gilead in spring 2015. EW has had tuition fees and a stipend paid by Gilead. AC received advisory board fees from Gilead Sciences plc and GSK/ViiV; speaker fees from Gilead and conferences bursaries from Gilead & Janssen. SM reports grants from the European Union H2020 scheme, EDCTP 2, the National Institute of Health Research and Gilead Sciences; other support from Gilead Sciences, and the Population Council Microbicide Advisory Board and is Chair of the Project Advisory Committee for USAID grant awarded to CONRAD to develop tenofovir-based products for use by women (non-financial).

**Patient consent for publication** Not required.

**Ethics approval** The study was reviewed and approved by London Bridge Research Ethics Committee (12/LO/1289). Written informed consent was obtained from all participants. The study protocol is available online (http://www.proud.mrc.ac.uk/about/study-protocol/).

**Provenance and peer review** Not commissioned; externally peer reviewed.

**Data availability statement** The PROUD data is held at MRC CTU at UCL, which encourages optimal use of data by employing a controlled access approach to data sharing, incorporating a transparent and robust system to review requests and provide secure data access consistent with the relevant ethics committee approvals. All requests for data are considered and can be initiated by contacting proud.mrcctu@ucl.ac.uk. Data from the current analysis are not available.

**ORCID iD**
Ada Miltz http://orcid.org/0000-0003-2771-7880

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
