## [Reviewer comments · BMJ Open]

ARTICLE DETAILS

TITLE (PROVISIONAL)	Prevalence and correlates of depressive symptoms among gay, bisexual and other men who have sex with men in the PROUD randomized clinical trial of HIV pre-exposure prophylaxis
AUTHORS	Miltz, Ada; Lampe, F; McCormack, Sheena; Dunn, David; White, Ellen; Rodger, Alison; Phillips, Andrew; Sherr, Lorraine; Sullivan, Ann; Reeves, Iain; Clarke, Amanda; Gafos, Mitzy

VERSION 1 – REVIEW

REVIEWER	Angela Bengtson Brown University, USA
REVIEW RETURNED	26-May-2019

GENERAL COMMENTS	This paper reports on the prevalence and correlates of depressive symptoms among gay, bisexual and other men who have sex with men in the PROUD Trial, which examined the use of immediate versus delayed PrEP initiation. This analysis helps to define the burden and correlates of depression in this important population for HIV prevention and is therefore an important contribution to the literature. I have several suggestions to hopefully improve the clarify and interpretation of the paper and some questions on the analysis it would be helpful for the authors to address. Minor Points • PHQ-9 not mentioned in the methods of the abstract, but referred to in the results. PHQ-9 is not spelled out before it's used in the abstract.• End of first paragraph and beginning of second paragraph in introduction are a bit repetitive and could be streamlined.• In the methods, part of the inclusion criteria is "male at birth". Presumably the authors mean male sex at birth?• Lines 56 and 57 have multiple acronyms that have not been previously defined. In addition, this sentence is rather vague. Either a description of the information available at the link provided or some examples of participant engagement activities (or both), is needed.• When discussing the measurement properties of the PHQ-9, please include the sensitivity and specificity of the measure to diagnose major depression.• Please clarify the following statement: across time-points [within trial arms?] using McNemar's χ^2 test to account for paired responses.• This sentence is a bit confusing to read: "The prevalence of depressive symptoms was slightly higher in the immediate than the deferred PrEP group at baseline and at subsequent time-points, but prevalence did not differ significantly between trial arms at months 12 and 24 (Figure 1a)." I recommend revising to focus on
---

	the magnitude of differences between the trial arms over time, rather than statistical significance.  • Again, some description of the magnitude of these changes on average would be helpful in the following sentence: “The within-person increase in prevalence observed from baseline to month-12 and from baseline to month-24 was significant (Figure 1b & Table 1a – 1c) but there was no difference from month-12 to month-24.” • The following sentence is rather vague: “Patient and public involvement with PROUD participants also highlighted the potential of greater familiarity with the study over time resulting in increased reporting of depressive symptoms at follow-up visits (http://www.proud.mrc.ac.uk/)”. Please indicate what you are referring to in the provided link. • Major Points  • The authors state that one of the key strengths of the paper is the collection of “longitudinal data on depression prevalence over time”; however, the cross sectional analyses don’t take advantage of the longitudinal structure of the data and no justification for doing a cross sectional analysis is given. Please provide a rationale for structuring the statistical analysis in this way. In addition, to address questions around directionality of effects, the authors should take advantage of the longitudinal data to examine whether factors at baseline help to predict depressive symptoms at 12 months and if factors at 12 months predict depressive symptoms at 24 months. • Concerns about missing data and attrition:  o Assuming all missing values are a 0 for the PHQ-9 and all other psychosocial measures is a very strong assumption and it’s not clear this is justified. It seems feasible that the authors could consult with the study team to determine if this assumption is reasonable (beyond simply looking at the data). In addition, a sensitivity analysis should be conducted examining how this assumption of missing=0 affects results (e.g. leave missing values missing). If missing data is a considerable issue, multiple imputation or other missing data methods should be considered in sensitivity analysis. o Please clarify what you mean by “A sensitivity analysis was undertaken excluding missing values when defining each variable”. How were missing values treated in the primary analysis? Unless each variable had a “missing data” category, then missing values would have been dropped in statistical models. Does this refer psychosocial measures that were missing and coded as 0 in the primary analysis? o Attrition is considerable over the follow up period, but is not addressed in your analysis. Presumably this may have been why the authors chose to do examine the data cross sectionally? The issue of attrition and how it affected decisions about how to analyze the data needs to be addressed clearly in the paper. Moreover, I would recommend considering methods to account for the attrition (such as inverse probability weighting) to try to address possibly informative attrition. • In general, much more information on the measures used in the analysis is needed in the Methods section and not just the footnotes of tables. This would also cut down on the number of footnotes in the tables, which are difficult to follow. For example:  o IPV, homophobia, alcohol use, relationship status – all need to be discussed and clarified in the methods. A definition of how each
--	--

	measure was defined and coded and indication of the validity of each measure is needed, particularly among GSMSM.  • Concerns about the statistical analysis:  o It would be helpful to clarify the goals of the statistical analysis. For example, if the goal is to understand whether some factors are associated with depressive symptoms, how did you decide which factors you wanted to explore and which factors would be considered confounders? Factors such as sexual identity and relationship status might also correlate with depressive symptoms. Therefore, some justification for what was considered a potential “correlate” and what was treated as a confounder is needed. o A lot of these measures could be correlated (particularly among the IPV measures) was collinearity evaluated? If so what were the results and how was it handled? o What is the justification for controlling for follow-up point in the GEE analysis? It’s not clear to me why this is done given that correlation between outcomes is already accounted for by the GEE model. • Suggestions for tables/figures:  o Table 1 needed to describe the study sample. This would also cut down on the amount of descriptive information in tables 2 and 3 and make them much clearer and easier to read. o Tables 1a-c are confusing and not well labeled. The titles should indicate that the tables are comparing participants who were enrolled at both baseline and 12 months for example and the proportion of men with depressive symptoms at each time point should be clear (for example in Table 1a, one has to calculate what 38/410 is to find the prevalence of depressive symptoms at baseline, among men included at both baseline and 12 months). Moreover, I don’t know if these tables are all that useful due to the strong suggestion of underreporting of depressive symptoms at baseline after April 2013 in your data. Rather, it may be more useful to present graphs demonstrating the prevalence of depressive symptoms, stratified by participants included at each time-point and by calendar time (e.g. pre April 2013 and post April 2013). o Figures 1a and 1b seem to include some of this information, but again I found confusing to understand. Some of the labels seem to be misplaced (for example, in Figure 1a the arrows at 24 months are not labeled) and it wasn’t clear to me what the arrows represented (a point estimate? An increase?). Figure 1a also is labeled as included men with data at all time points, but the N’s change between each time point, suggesting what’s actually graphed is the prevalence among men with data at any time point. o Table 3 – the unadjusted and adjusted estimates should be among the same study population. Making this adjustment would also cut down on the footnotes in the table and make it easier to read.
--	---

REVIEWER	Christopher F. Drescher Augusta University, U.S.
REVIEW RETURNED	08-Aug-2019

GENERAL COMMENTS	The current study describes an analysis of depression and its correlates within an RCT of PrEP for gay, bisexual, and other men who have sex with men in the UK. It included data from baseline, 12-month, and 24-month follow up. Depression increased from baseline to 12 and 24 months, possibly as a result of methodological change that may have led to underreporting of depression at baseline. Several significant correlates of
---

	depression were elucidated, including interpersonal violence, methamphetamine use, internalized homophobia, and low levels of outness. The manuscript is generally well written and concise. It addresses an important topic, has an adequate sample size, and focuses on a region (UK) where there has been less research on the topic as compared to other countries, such as the US. That said I believe the manuscript could be improved in several ways. I have two overarching comments and will address my other comments by manuscript section. The most significant aspect that the manuscript is missing is a coherent theoretical framework for the various correlates that are explored. Clearly, this was not a primary analysis from the PROUD study and it is understandable that the research team had to use the measures that were available, even when they did not address the specific construct as fully as they would have liked (e.g., chemsex). However, it would be very helpful to the reader to have an overarching framework introduced in the introduction and revisited in the discussion to. Minority Stress Theory, mentioned near the end of the discussion would be a natural choice. However, other approaches, such as syndemic theory, could also be applied. Regardless of the specific framework, some framework is needed to help the reader integrate previous research findings with the various findings from the study. This would likely lead to some expansion of the introduction and discussion, which are currently brief. My second overarching point is that very little is said in terms of the implications of this study. Beyond a single line in the conclusions section, there is no indication of what this study means in terms of future research or clinical implications. I implore the authors to engage in more discussion about the potential implications of this study. Comments by section: Strengths and limitations (bullet points before introduction)  • Please define the term chemsex as many readers may not be familiar with it. It is defined in the manuscript but should be defined here as well. Introduction  • The first paragraph highlights various aspects of depression. Depression is such a broad topic that it is difficult to review briefly and the paragraph as currently written feels unfocused as it starts with very general information about depression and then notes several specific links related to study variables. I think this paragraph could be improved by choosing a guiding theory as already mentioned above to focus on how the findings all fit together. • A citation should be provided for the previous PROUD study that examined the link between depression and sexual risk behaviors. It would be helpful to briefly mention the results of the study in terms of a link between depression and sexual risk behaviors, if one existed in the previous publication. • The concept of chemsex should be introduced somewhere in the introduction as it is included in the study.
--	---

	 • Although the goals of study are defined, they are very broad and the introduction should include more specific study hypotheses regarding the expected findings. Methods  • Most of the information about the measures is included in footnotes to the tables. I would suggest including this information in the methods section and simplifying the extensive footnotes for the tables. • Regarding the measure of internalized homophobia, why were only 8 items used and how were those eight items selected? • The measure of outness needs to be included in the methods section as it is included in the results. Discussion  • The sentence that starts “Evidence from a recent meta-analysis of mostly U.S....” needs to clarify IPV victimization, perpetration, or both are associated with depression. • The strengths and limitations should acknowledge some additional limitations including limited information on economic status and childhood traumatic experiences (since the authors make the point that this is linked to depression). Also, the change in methods that may have affected the reporting of depressive symptoms is a significant limitation. • Add future research and practical implications, as discussed above. Figures  • I suggest removing Figures 1a and 1b as they overlap greatly with the analysis that is already described in text. If they remain, they need to be reformatted as text is overlapping in the upper left-hand corners and some of the percentages are missing or incorrect.
--	---

VERSION 1 – AUTHOR RESPONSE

Reviewer#1:
Minor points

1. PHQ-9 not mentioned in the methods of the abstract, but referred to in the results. PHQ-9 is not spelled out before it's used in the abstract.
Response: We thank the reviewer for this comment. PHQ-9 is now mentioned (and spelled out) in the methods of the abstract.
2. End of first paragraph and beginning of second paragraph in introduction are a bit repetitive and could be streamlined.
Response: The first paragraph has been removed (page 3).
3. In the methods, part of the inclusion criteria is “male at birth”. Presumably the authors mean male sex at birth?
Response: We thank the reviewer for this comment. This has been corrected (page 4, paragraph 2).

4. Lines 56 and 57 have multiple acronyms that have not been previously defined. In addition, this sentence is rather vague. Either a description of the information available at the link provided or some examples of participant engagement activities (or both), is needed.

Response: These acronyms have been removed, and a brief description of participant engagement activities is now provided (page 5, paragraph 2).

5. When discussing the measurement properties of the PHQ-9, please include the sensitivity and specificity of the measure to diagnose major depression.

Response: This is now provided (page 6, paragraph 1): "In a U.S. validation study of primary care and obstetrics/gynaecology patients, the vast majority of patients (93%) with no depressive disorder diagnosed by a (blinded) mental health professional had a PHQ-9 score less than 10, while 88% with a major depression diagnosis had scores of 10 or greater 20."

6. Please clarify the following statement: across time-points [within trial arms?] using McNemar's χ^2 test to account for paired responses.

Response: This statement has now been clarified (page 8, paragraph 3): "across time-points (trial arms combined)".

7. This sentence is a bit confusing to read: "The prevalence of depressive symptoms was slightly higher in the immediate than the deferred PrEP group at baseline and at subsequent time-points, but prevalence did not differ significantly between trial arms at months 12 and 24 (Figure 1a)." I recommend revising to focus on the magnitude of differences between the trial arms over time, rather than statistical significance.

Response: This paragraph has been revised (page 10, paragraph 2): "The prevalence of depressive symptoms was slightly higher in the immediate than the deferred PrEP group at baseline (10.6% versus 7.5%) and at subsequent time-points (16.7% versus 11.9% at month-12 and 16.5% versus 11.9% at month-24). Differences were not statistically significant at any time-point ($p \geq 0.2$). The within-person increase in prevalence from 9.3% at baseline to 14.4% at month-12 (among 410 men) and from 9.6% at baseline to 14.4% at month-24 (among 333 men) was significant ($p \leq 0.04$), but there was no difference from 14.0% at month-12 to 14.7% at month-24 (among 307 men)."

8. Again, some description of the magnitude of these changes on average would be helpful in the following sentence: "The within-person increase in prevalence observed from baseline to month-12 and from baseline to month-24 was significant (Figure 1b & Table 1a – 1c) but there was no difference from month12 to month-24."

Response: Please see our response to question 7 above.

9. The following sentence is rather vague: "Patient and public involvement with PROUD participants also highlighted the potential of greater familiarity with the study over time resulting in increased reporting of depressive symptoms at follow-up visits ([https://eur01.safelinks.protection.outlook.com/?url=http%3A%2F%2Fwww.proud.mrc.ac.uk%2F&](https://eur01.safelinks.protection.outlook.com/?url=http%3A%2F%2Fwww.proud.mrc.ac.uk%2F&data=02%7C01%7C%7C5ce50370f63b45081e5908d73511ca22%7C1faf88fea9984c5b93c9210a11d9a5c2%7C0%7C0%7C637036222832132183&sd=QuhPxMFKpy%2BKhjTdl8Kck41PWy0TBNzjY9I7zoDV2Q%3D&reserved=0)

[;data=02%7C01%7C%7C5ce50370f63b45081e5908d73511ca22%7C1faf88fea9984c5b93c9210a11d9a](https://eur01.safelinks.protection.outlook.com/?url=http%3A%2F%2Fwww.proud.mrc.ac.uk%2F&data=02%7C01%7C%7C5ce50370f63b45081e5908d73511ca22%7C1faf88fea9984c5b93c9210a11d9a5c2%7C0%7C0%7C637036222832132183&sd=QuhPxMFKpy%2BKhjTdl8Kck41PWy0TBNzjY9I7zoDV2Q%3D&reserved=0)

[5c2%7C0%7C0%7C637036222832132183&sd=QuhPxMFKpy%2BKhjTdl8Kck41PWy0TBNzjY](https://eur01.safelinks.protection.outlook.com/?url=http%3A%2F%2Fwww.proud.mrc.ac.uk%2F&data=02%7C01%7C%7C5ce50370f63b45081e5908d73511ca22%7C1faf88fea9984c5b93c9210a11d9a5c2%7C0%7C0%7C637036222832132183&sd=QuhPxMFKpy%2BKhjTdl8Kck41PWy0TBNzjY9I7zoDV2Q%3D&reserved=0)

[9I7zoDV2Q%3D&reserved=0">9I7zoDV2Q%3D&reserved=0"\)](https://eur01.safelinks.protection.outlook.com/?url=http%3A%2F%2Fwww.proud.mrc.ac.uk%2F&data=02%7C01%7C%7C5ce50370f63b45081e5908d73511ca22%7C1faf88fea9984c5b93c9210a11d9a5c2%7C0%7C0%7C637036222832132183&sd=QuhPxMFKpy%2BKhjTdl8Kck41PWy0TBNzjY9I7zoDV2Q%3D&reserved=0). Please indicate what you are referring to in the provided link.

Response: This link has been removed. A brief description of this report has now been provided (page

12, paragraph 4): "Patient and public involvement activities with a small group of PROUD participants (N=13), in the form of a short qualitative survey, provided some participant perspectives on the change in depressive symptoms over time. Some highlighted the possibility that greater familiarity

with the questionnaire and study staff over time may have resulted in increased reporting of depressive symptoms at follow-up visits. Additionally, partaking in a clinical trial was thought to be one of the processes that helps individuals recognize/acknowledge their own depression in an environment where access to support is provided. Other suggested explanations for an actual increase in depressive symptoms included the possibility of greater socio-economic hardship in a period of national financial instability, risk taking behaviours in relation to CAS and chemsex, or PrEP-related stigma. None of the 13 participants specifically theorized that engagement in chemsex increased over the study, but chemsex was a common feature in discourse on depression. Although these participants were informed of the lack of difference between the immediate and deferred PrEP groups with regards to depression prevalence, some men cited that PrEP related side-effects and stigma, as well as fear of loss of access to PrEP after study completion, may have increased over time.”

Major points

1. “The authors state that one of the key strengths of the paper is the collection of “longitudinal data on depression prevalence over time”; however, the cross sectional analyses don’t take advantage of the longitudinal structure of the data and no justification for doing a cross sectional analysis is given. Please provide a rationale for structuring the statistical analysis in this way. In addition, to address questions around directionality of effects, the authors should take advantage of the longitudinal data to examine whether factors at baseline help to predict depressive symptoms at 12 months and if factors at 12 months predict depressive symptoms at 24 months.”

Response: Thank you for this comment. Longitudinal analyses investigating predictors of depression incidence are complicated by the fact that incident cases of depression may reflect new or recurring episodes. This may affect the temporality of associations found. Data on previous diagnoses and/or treatment for depression is therefore important when addressing depression incidence. We identified men who appeared to have no history of depression based on PROUD data; PHQ-9<10 at baseline and month-12 and no antidepressant use, as captured at baseline only. Analysing data from these men might increase the chances that ‘incident’ cases of depression (PHQ-9≥10) observed at month-24, were in fact new. Using 12-month data as the baseline for longitudinal analysis would also address the issue that most psychosocial factors were not asked about at the baseline visit. We found that of the 233 men who appeared to have no history of depression by month-12 in PROUD, 22 reported clinically significant symptoms at month-24. This longitudinal analysis does not have sufficient statistical power to detect associations. The data were therefore treated as cross-sectional. Generalized estimating equations (GEEs) were used to combine data from the two follow-up time-points to boost the power of the analysis. The following has been included in the ‘Statistical analysis’ section of the Methods (page 9, paragraph 1); “A longitudinal analysis was not feasible as the number of ‘incident’ cases of PHQ-9 depression was low; of the 233 men who appeared to have no history of depression based on PROUD data (PHQ-9<10 at baseline and month-12 and no antidepressant use, as captured at baseline only), 22 reported clinically significant symptoms (PHQ-9≥10) at month-24.”. It has been reiterated in the ‘Strengths and limitations’ section that there is a need to conduct a longitudinal study designed to address depression among GBMSM.

Concerns about missing data and attrition:

2. “Assuming all missing values are a 0 for the PHQ-9 and all other psychosocial measures is a very strong assumption and it’s not clear this is justified. It seems feasible that the authors could consult with the study team to determine if this assumption is reasonable (beyond simply looking at the data). In addition, a sensitivity analysis should be conducted examining how this assumption of missing=0 affects results (e.g. leave missing values missing). If missing data is a considerable issue, multiple imputation or other missing data methods should be considered in sensitivity analysis.”

Response: We thank the reviewer for this comment. Given that the proportion of missing responses was <5% for each measure (at each time-point), imputation techniques to deal with missingness were

not considered necessary. For a number of variables including depressive symptoms on PHQ-9, missing responses were considered to indicate that an event did not occur. The rationale being that the proportion of people who had a missing response to all nine questions on the PHQ-9 was very low (3% at baseline and 1% at month-12 and month-24). There appeared to be a common response pattern in which people only ticked PHQ-9 symptoms when they had experienced it and did not tick the response 'Not at all' to all those symptoms they did not have. This same rationale extended to measures of drinking, drug use, sexualized drug use, internalized homophobia, and IPV. Given the very low proportion of participants with missing values on socio-demographic factors and the relatively small size of the sample, a similar method of dealing with 'missingness' was utilized for these measures where appropriate. Missing responses were considered to indicate a lower level of educational attainment (i.e. no university degree), no ongoing relationship, and no current employment. We appreciate the need to investigate whether removing non-respondents from the analysis has any impact on the findings. We hope it is satisfactory that we have examined all analyses excluding participants with missing responses and found that the model estimates were almost identical. The study team were in agreement that given no difference in findings was observed, it is appropriate to present estimates from models with missingness included. The following rationale has been added to the Statistical analysis section of the Methods (page 9, paragraph 2): "For each measure and at each time-point, the proportion of missing responses was <5%. For psychosocial measures (depression, drinking, drug use, sexualized drug use, internalized homophobia, and IPV), missing responses were considered to indicate that the event did not occur. There appeared to be a common response pattern in which only those experiences that had occurred were ticked on the questionnaire. Given the very low proportion of missing responses, missing values were considered to indicate a lower level of educational attainment (i.e. no university degree), no ongoing relationship, and no current employment. A sensitivity analysis was undertaken excluding missing values when defining each variable. The findings were very similar to the main analysis."

3. "Please clarify what you mean by "A sensitivity analysis was undertaken excluding missing values when defining each variable". How were missing values treated in the primary analysis? Unless each variable had a "missing data" category, then missing values would have been dropped in statistical models. Does this refer psychosocial measures that were missing and coded as 0 in the primary analysis?" Response: Please see our response to question 2 (major points) above.

4. "Attrition is considerable over the follow up period, but is not addressed in your analysis. Presumably this may have been why the authors chose to do examine the data cross sectionally? The issue of attrition and how it affected decisions about how to analyze the data needs to be addressed clearly in the paper. Moreover, I would recommend considering methods to account for the attrition (such as inverse probability weighting) to try to address possibly informative attrition." Response: Issues surrounding attrition have now been addressed in the 'Strengths and limitations' section of the Discussion (page 15, paragraph 2): "Given the relatively small sample size of the PROUD trial, the examination of factors associated with depression was based on cross-sectional analysis, utilising all available observations. Some participants were not included in the GEE analysis of psychosocial correlates due to missing questionnaires at months 12 and 24. However, men with and without depressive symptoms (PHQ-9 \geq 10) at baseline had similar proportions completing the 12-month questionnaire (22.5% vs. 24.2%; χ^2 test $p=0.780$) and those with and without depressive symptoms at the 12-month questionnaire had similar proportions completing the 24-month questionnaire (27.1% vs. 24.8%; χ^2 test $p=0.702$). When stratifying by trial arm, there remained no difference between men with and without depressive symptoms in terms of proportion completing follow-up questionnaires. Even after including repeated observations in GEE models, the analysis may have lacked power to accurately detect the presence of some associations."

5. “In general, much more information on the measures used in the analysis is needed in the Methods section and not just the footnotes of tables. This would also cut down on the number of footnotes in the tables, which are difficult to follow. For example: IPV, homophobia, alcohol use, relationship status – all need to be discussed and clarified in the methods. A definition of how each measure was defined and coded and indication of the validity of each measure is needed, particularly among GSMSM.”

Response: The psychosocial measures investigated in this paper are now described in detail in the Methods section (pages 5 to 8). This includes, depression, intimate partner violence, age at anal sex debut, recreational drug use and alcohol use, and internalized homophobia. The internal reliability of scales within GBMSM samples has been reported where available.

Concerns about the statistical analysis:

6. “It would be helpful to clarify the goals of the statistical analysis. For example, if the goal is to understand whether some factors are associated with depressive symptoms, how did you decide which factors you wanted to explore and which factors would be considered confounders? Factors such as sexual identity and relationship status might also correlate with depressive symptoms. Therefore, some justification for what was considered a potential “correlate” and what was treated as a confounder is needed.”

Response: We thank the reviewer for this comment. The Introduction has been revised such that the theoretical models (minority stress theory and syndemic theory) used to guide the selection of variables in this analysis are now described (page 3). The aims have been adapted accordingly (page 4, paragraph 3): “The aim is to: (i) present the prevalence of depressive symptoms at baseline and follow-up, examining changes in prevalence over time and (ii) drawing on minority stress theory and syndemic theory, investigate the association of demographic, socio-economic and psychosocial factors with depressive symptoms.” A section has been added to the methods ‘Hypothesized relationships and selection of variables’ (page 5, paragraph 3), which describes the causal relationships hypothesized with depression in this analysis, based on theoretical and Epidemiological evidence. The measures used in the analysis are described in the next section. Finally, in the ‘Statistical analysis’ section of the methods (page 7, paragraph 4), it is now stated that seven socio-demographic factors, which were assumed not to be on hypothesized causal pathways from socio-economic and psychosocial factors investigated to depression, are adjusted for: “Seven socio-demographic factors assumed not to be on hypothesized causal pathways were adjusted for: Age (<25, 25-29, 30-39, or 40+), born in the UK (yes or no/missing), sexual identity (gay or bisexual/straight), university education (yes or no/missing), ongoing relationship (yes or no/missing), London study clinic site (yes or no), and calendar time (year and calendar quarter).”

7. A lot of these measures could be correlated (particularly among the IPV measures). Was collinearity evaluated? If so what were the results and how was it handled?

Response: For each factor, separate socio-demographic adjusted models were constructed. Therefore, only one measure of IPV (either lifetime IPV victimization or lifetime IPV perpetration) was included in a model. It was not considered necessary to investigate possible collinearity since only factors that were considered to be measuring separate constructs were included in the same model. This point has now been clarified in the paper (see page 7, paragraph 4).

8. What is the justification for controlling for follow-up point in the GEE analysis? It’s not clear to me why this is done given that correlation between outcomes is already accounted for by the GEE model.

Response: In our combined analysis, individuals reported data for the same measure twice (i.e. for IPV at month-12 and for IPV at month-24). GEEs take into account the correlation between these observations. We adjusted for time-point (month-12 or month-24) in order to control for any time-specific effects that impact on depression as well as their risk factors. For instance, if most

participants at month-12 reported no IPV and no depression but at month-24 reported IPV and depression, an association between IPV and depression would be observed in GEE, but there also appears to be a time-specific effect i.e. something (possibly PrEP related stigma for instance) at month-24 may have influenced experiences of psychosocial factors. Therefore, in order to make sure it is not unmeasured factors associated with time-point that is creating the observed relationship, time-point needs to be adjusted for.

Suggestions for tables/figures:

9. Table 1 needed to describe the study sample. This would also cut down on the amount of descriptive information in tables 2 and 3 and make them much clearer and easier to read.

Response: Table 1 now describes the study sample (page 19).

10. "Tables 1a-c are confusing and not well labelled. The titles should indicate that the tables are comparing participants who were enrolled at both baseline and 12 months for example and the proportion of men with depressive symptoms at each time point should be clear (for example in Table 1a, one has to calculate what 38/410 is to find the prevalence of depressive symptoms at baseline, among men included at both baseline and 12 months). Moreover, I don't know if these tables are all that useful due to the strong suggestion of underreporting of depressive symptoms at baseline after April 2013 in your data. Rather, it may be more useful to present graphs demonstrating the prevalence of depressive symptoms, stratified by participants included at each time-point and by calendar time (e.g. pre April 2013 and post April 2013)."

Response: Tables 1a to 1c have been removed. This data has now been clearly described in the text (page

10, paragraph 1): "In terms of changes in individual depression status between time-points, among the 410 men who completed the baseline and 12-month questionnaire, 38 men reported depressive symptoms (PHQ-9 \geq 10) at baseline, 22 (57.9%) of whom continued to report this measure of depression at month-12. Of the 372 men who did not report depressive symptoms at baseline, 37 (10.0%) reported it at month-12. Among the 333 men who completed the baseline and 24-month questionnaire, 32 men who reported depressive symptoms (PHQ-9 \geq 10) at baseline, 11 (34.4%) of whom continued to report this measure of depression at month-24. Of the 301 men who did not report depressive symptoms at baseline, 37 (12.3%) reported it at month-24. Finally, among the 307 men who completed both the 12- and 24month questionnaire, of 43 men who reported depressive symptoms (PHQ-9 \geq 10) at month-12, 16 (37.2%) continued to report this measure of depression at month-24. Of the 264 men who did not report depression at month-12, 29 (11.0%) reported it at month-24." The prevalence of depressive symptoms at baseline pre April 2013 and post April 2013, and the change in prevalence observed for both groups at month-12, is presented and discussed on page 11 under the section 'Exploration of changes in depression prevalence'. Therefore, we believe it is not necessary to reproduce this data in a figure, and hope the reviewer agrees.

11. "Figures 1a and 1b seem to include some of this information, but again I found confusing to understand. Some of the labels seem to be misplaced (for example, in Figure 1a the arrows at 24 months are not labeled) and it wasn't clear to me what the arrows represented (a point estimate? An increase?). Figure 1a also is labeled as included men with data at all time-points, but the N's change between each time point, suggesting what's actually graphed is the prevalence among men with data at any time point." Response: Figures 1a and 1b have now been removed (also requested by reviewer 2, see the final point, point 1 under 'Figures'). This data has now been clearly described in the text (page 10, paragraph 2): "The prevalence of depressive symptoms was slightly higher in the immediate than the deferred PrEP group at baseline (10.6% versus 7.5%) and at subsequent time-points (16.7% versus 11.9% at month-12; 16.5% versus 11.9% at month-24). Differences between randomised groups were not statistically significant at any time-point ($p\geq 0.2$). The within-person increase in prevalence from 9.3% at baseline to 14.4% at month-12 (among 410 men) and from 9.6%

at baseline to 14.4% at month-24 (among 333 men) was significant ($p \leq 0.04$), but there was no difference from 14.0% at month-12 to 14.7% at month-24 (among 307 men)."

12. "Table 3 – the unadjusted and adjusted estimates should be among the same study population. Making this adjustment would also cut down on the footnotes in the table and make it easier to read." Response: Unadjusted associations were assessed among all participants with available data on measures under investigation. We believe this is the best approach given the small sample size and our desire to include all data points. The proportion of missing data on adjusted for socio-demographic factors was very small (<5%), therefore the denominators for unadjusted and adjusted analysis were very similar (i.e. only differed by 3-4 participants). The footnotes have been reduced, as measures are now described in the Methods section.

Reviewer#2:
Overarching comments

1. "The most significant aspect that the manuscript is missing is a coherent theoretical framework for the various correlates that are explored. Clearly, this was not a primary analysis from the PROUD study and it is understandable that the research team had to use the measures that were available, even when they did not address the specific construct as fully as they would have liked (e.g., chemsex). However, it would be very helpful to the reader to have an overarching framework introduced in the introduction and revisited in the discussion to. Minority Stress Theory, mentioned near the end of the discussion would be a natural choice. However, other approaches, such as syndemic theory, could also be applied. Regardless of the specific framework, some framework is needed to help the reader integrate previous research findings with the various findings from the study. This would likely lead to some expansion of the introduction and discussion, which are currently brief."

Response: We thank the reviewer for this comment. Please see our above response to reviewer 1 (major points, point 6). The Introduction has been revised such that the theoretical models (minority stress theory and syndemic theory) used to guide the selection of variables in this analysis are now described (page 3). The aims have been adapted accordingly (page 4, paragraph 3): "The aim is to: (i) present the prevalence of depressive symptoms at baseline and follow-up, examining changes in prevalence over time and (ii) drawing on minority stress theory and syndemic theory, investigate the association of demographic, socio-economic and psychosocial factors with depressive symptoms." A section has been added to the methods 'Hypothesized relationships and selection of variables' (page 5, paragraph 3), which describes the causal relationships hypothesized with depression in this analysis, based on theoretical and Epidemiological evidence. The Discussion has also been expanded (page 14, paragraphs 2 and 3).

2. "My second overarching point is that very little is said in terms of the implications of this study. Beyond a single line in the conclusions section, there is no indication of what this study means in terms of future research or clinical implications. I implore the authors to engage in more discussion about the potential implications of this study."

Response: We thank the reviewer for this comment. The 'Conclusion' section has been expanded to include a number of study implications in terms of possible interventions and future research (page 16): "In the context of interviewer administered surveys or surveys in which interviewers/study staff are made aware of individual responses, participants may be reluctant to disclose information on depressive symptomatology. This may be particularly true in clinical trial settings that are subject to strict inclusion/exclusion criteria, as eligible participants may perceive the disclosure of current depressive symptoms as prohibiting study inclusion. Researchers should be aware of the potential limitations to reporting depression prevalence under these circumstances."

Our findings add to the growing body of research in high-income countries suggesting a high concomitant burden of sexual minority related stress, substance use, and IPV with depression among GBMSM. Since these psychosocial measures are relevant to GBMSM at high-risk of HIV and seeking PrEP, it is recommended that training on awareness and enquiry about psychosocial issues should be enhanced in sexual health services. This should include referral to substance use services and culturally appropriate structures set up to address IPV among same-sex male couples. Furthermore, findings suggest that new interventions addressing sexual minority related stresses are needed in order to foster greater community affiliation and support, and curtail stress. Intervening with sexual minority youths in schools and/or in LGBT+ youth groups may prevent the internalizing of homophobic attitudes and the onset of mental health problems. Although it is important to be cognizant of the unique forms of stress experienced by sexual minority men, findings from PROUD also emphasise the link between socioeconomic disadvantage and poor mental health. Therefore, emphasis on alleviating economic hardship together with efforts to promote self-acceptance and pride towards one's sexual orientation is recommended when addressing depression among sexual minority men.

In many countries, sexual minorities are being accepted and celebrated more so than ever in history 40-43. This social climate provides a solid platform from which to launch extensive efforts to eliminate homophobic discrimination in society at large. Such a structural change is likely to be profoundly beneficial to the health and wellbeing of sexual minority people. Efforts may include national educational campaigns and specialised training for health care providers on homophobia and heterosexism." Comments by section:

Strengths and limitations (bullet points before introduction)

1. Please define the term chemsex as many readers may not be familiar with it. It is defined in the manuscript but should be defined here as well.

Response: The term chemsex has now been defined (page 3).

Introduction:

1. The first paragraph highlights various aspects of depression. Depression is such a broad topic that it is difficult to review briefly and the paragraph as currently written feels unfocused as it starts with very general information about depression and then notes several specific links related to study variables. I think this paragraph could be improved by choosing a guiding theory as already mentioned above to focus on how the findings all fit together.

Response: The introduction has been rewritten. This section is now focused upon minority stress theory and syndemic theory, as frameworks to understand factors associated with depression prevalence among sexual minority men (page 3, paragraph 1): "There is consistent evidence that depression prevalence is elevated among gay and bisexual men compared to their heterosexual counterparts in high-income countries 1. Two theoretical models have put forward explanations for the elevated prevalence observed among sexual minorities: minority stress theory and syndemic theory. Minority stress theory describes the psychosocial consequences of being in continual conflict with a discriminatory and heteronormative social environment. Perpetual negative feedback from others is thought to lead to a process of selfstigmatization termed internalized homophobia, whereby antigay social values/attitudes are directed towards the self. Internalized homophobia often results in deep conflict and poor self-regard with negative consequences such as pervasive expectations of rejection in one's life and concealment of one's sexual identity. Concealment adds to mental distress by disallowing individuals to affiliate with people of the same sexual identity 2. Two U.S. studies of gay, bisexual and other men who have sex with men (GBMSM) found strong associations between perceived discrimination on the basis of sexual orientation and depressive symptoms 3 4. Internalized homophobia may explain this association. Syndemic theory proposes that concomitant widespread use of recreational drugs and experiences of intimate partner violence (IPV) among sexual minority men may be to blame for elevated levels of depression, as causal, and bidirectional, relationships are

postulated between drug use, IPV, and depression. The synergistic interaction of these co-occurring factors is thought to result in an exaggerated risk of poor health outcomes 5. A number of U.S. studies have found strong associations of recreational drug use 3 6 7 and IPV 8 with depressive symptoms. Recently, studies have also found a link between chemsex and depression 9-11.

Chemsex is a cultural phenomenon among a sub-group of gay identified men, which was first described in the UK 12. It is the intentional use of psychoactive substances (usually one or more of mephedrone, gamma-hydroxybutyrate/gamma-butyrolactone [GHB/GBL] and methamphetamine) to stimulate sexual arousal, facilitate different sexual practices, and prolong sexual episodes.”

2. A citation should be provided for the previous PROUD study that examined the link between depression and sexual risk behaviors. It would be helpful to briefly mention the results of the study in terms of a link between depression and sexual risk behaviors, if one existed in the previous publication. Response: This has now been provided (page 12, paragraph 2). The results of this study are briefly described.

3. “The concept of chemsex should be introduced somewhere in the introduction as it is included in the study.”

Response: The concept of chemsex is now introduced in the Introduction (page 4, paragraph 1).

4. “Although the goals of study are defined, they are very broad and the introduction should include more specific study hypotheses regarding the expected findings.”

Response: The aims have been adapted such that it is now clear that the analysis draws on minority stress theory and syndemic theory (page 3, paragraph 1). A section entitled ‘Hypothesized relationships and selection of variables’ (page 5, paragraph 3), has been added to the Methods, which describes the causal relationships hypothesized with depression in this analysis, based on theoretical and Epidemiological evidence.

Methods:

1. “Most of the information about the measures is included in footnotes to the tables. I would suggest including this information in the methods section and simplifying the extensive footnotes for the tables.” Response: The psychosocial measures investigated in this paper are now described in detail in the Methods section (pages 5 to 8). This includes, depression, intimate partner violence, age at anal sex debut, recreational drug use and alcohol use, and internalized homophobia. The footnotes to the tables have been removed.

2. “Regarding the measure of internalized homophobia, why were only 8 items used and how were those eight items selected?”

Response: In order to reduce the length of the survey, only eight questions surrounding attitudes to gay sexuality were asked in PROUD. Five of the ten questions (with high factor loadings) that form the largest sub-scale measuring attitudes to ‘Public identification as gay’ were used in PROUD. Two questions were also used from the sub-scale ‘Social comfort with gay men’, and one question was used from the sub-scale ‘Moral and religious acceptability of being gay’. These questions were considered to be representative of the construct of internalized homophobia.

3. “The measure of outness needs to be included in the methods section as it is included in the results.” Response: The measure of ‘outness’ is now described in the Methods (page 8, paragraph 2).

Discussion:

1. “The sentence that starts “Evidence from a recent meta-analysis of mostly U.S....” needs to clarify IPV victimization, perpetration, or both are associated with depression.”

Response: This has been clarified (page 14, paragraph 1): “Evidence from a recent meta-analysis of mostly U.S. studies suggests that exposure to any kind of IPV victimization is associated with increased odds of depressive symptoms...”

2. “The strengths and limitations should acknowledge some additional limitations including limited information on economic status and childhood traumatic experiences (since the authors make the point that this is linked to depression). Also, the change in methods that may have affected the reporting of depressive symptoms is a significant limitation.”

Response: These points have now been acknowledged in the ‘Strengths and limitations’ section of the

Discussion (page 15): “This is the first UK study to collect data on depression prevalence over time among GBMSM, and investigate associations with a range of psychosocial factors. A change in methodology part way through baseline data collection may have affected the reporting of depressive symptoms, limiting the validity of findings at baseline. Given the relatively small sample size of the PROUD trial, the examination of factors associated with depression was based on cross-sectional analysis, utilising all available observations. Some participants were not included in the GEE analysis of psychosocial correlates due to missing questionnaires at months 12 and 24. However, men with and without depressive symptoms (PHQ-9 \geq 10) at baseline had similar proportions completing the 12-month questionnaire (22.5% vs. 24.2%; χ^2 test $p=0.780$) and those with and without depressive symptoms at the 12-month questionnaire had similar proportions completing the 24-month questionnaire (27.1% vs.

24.8%; χ^2 test $p=0.702$). When stratifying by trial arm, there remained no difference between men with and without depressive symptoms in terms of proportion completing follow-up questionnaires. Even after including repeated observations in GEE models, the analysis may have lacked power to accurately detect the presence of some associations. Furthermore, the direction of any causality may be unclear for some associations. For this reason, as well as the possibility of unmeasured confounders, inferences regarding causal factors for depression are limited. Although chemsex drug use was investigated in this study, the cultural phenomenon of chemsex was not explicitly measured, as participants were not asked whether these drugs were taken to enhance sexual experiences.

There have been no longitudinal studies that are designed and powered to investigate depression among GBMSM. Future longitudinal analyses would be useful in investigating the suggested causal pathways in this paper. Such a study would require careful consideration in how to measure incident depression, as new symptoms may indicate first episode or recurrent depression. Therefore, capturing data on previous diagnoses and treatment is important. Following participants from a young age is likely to yield the most accurate data with regards to depression incidence. Finally, there is a need to examine associations with measures of family and community support, income, and abuse in childhood, as well as the role of these factors in relationships investigated in this paper.”

3. “Add future research and practical implications, as discussed above.”

Response: Study implications have been included (page 16). Please see our response above (overarching comments, point 2).

Figures:

1. “I suggest removing Figures 1a and 1b as they overlap greatly with the analysis that is already described in text. If they remain, they need to be reformatted as text is overlapping in the upper left-hand corners and some of the percentages are missing or incorrect.” Response: Figures 1a and 1b have been removed.

VERSION 2 – REVIEW

REVIEWER	Christopher Drescher Augusta University United States of America
REVIEW RETURNED	25-Oct-2019

GENERAL COMMENTS	I felt that the authors overall addressed my concerns expressed during the initial review and that the manuscript is overall greatly improved. I have some very minor recommendations regarding revision: - Break up the first paragraph into at least two paragraphs so that minority stress and syndemic theory have their own sections. - In the measures section, give each measure its own subsection (e.g., internalized homophobia an concealment of sexual identity should have their own subsections).
---

VERSION 2 – AUTHOR RESPONSE

Reviewer#2:

1. Break up the first paragraph into at least two paragraphs so that minority stress and syndemic theory have their own sections.

Response: We thank the reviewer for this comment. We have now split the first paragraph into two paragraphs, the first introducing minority stress theory and the second, syndemic theory.

2. In the measures section, give each measure its own subsection (e.g., internalized homophobia an concealment of sexual identity should have their own subsections).

Response: Each measure now has its own subsection (pages 7 to 8).